

# Rapid estimation of soil water content based on hyperspectral reflectance combined with continuous wavelet transform, feature extraction, and extreme learning machine

Shaomin Chen[1,2,*], Jiachen Gao[1,*], Fangchuan Lou[1], Yunfei Tuo[3], Shuai Tan[1,2], Yuyang Shan[4], Lihua Luo[5], Zhilin Xu[1], Zhengfu Zhang[1] and Xiangyu Huang[1]

[1] Faculty of Modern Agricultural Engineering, Kunming University of Science and Technology, Kunming, China
[2] Yunnan Provincial Field Scientific Observation and Research Station on Water-Soil-Crop System in Seasonal Arid Region, Kunming University of Science and Technology, Kunming, China
[3] Ecology and Environment Department, Southwest Forestry University, Kunming, China
[4] State Key Laboratory of Eco-Hydraulics in Northwest Arid Region of China, Xi'an University of Technology, Xi'an, China
[5] Yunnan Institute of Water and Hydropower Engineering Investigation and Design, Co., LTD, Kunming, China
[*] These authors contributed equally to this work.

Corresponding authors
Shuai Tan, tans90@163.com
Yuyang Shan, syy031@126.com

## ABSTRACT

**Background**. Soil water content is one of the critical indicators in agricultural systems. Visible/near-infrared hyperspectral remote sensing is an effective method for soil water estimation. However, noise removal from massive spectral datasets and effective feature extraction are challenges for achieving accurate soil water estimation using this technology.

**Methods**. This study proposes a method for hyperspectral remote sensing soil water content estimation based on a combination of continuous wavelet transform (CWT) and competitive adaptive reweighted sampling (CARS). Hyperspectral data were collected from soil samples with different water contents prepared in the laboratory. CWT, with two wavelet basis functions (mexh and gaus2), was used to pre-process the hyperspectral reflectance to eliminate noise interference. The correlation analysis was conducted between soil water content and wavelet coefficients at ten scales. The feature variables were extracted from these wavelet coefficients using the CARS method and used as input variables to build linear and non-linear models, specifically partial least squares (PLSR) and extreme learning machine (ELM), to estimate soil water content.

**Results**. The results showed that the correlation between wavelet coefficients and soil water content decreased as the decomposition scale increased. The corresponding bands of the extracted wavelet coefficients were mainly distributed in the near-infrared region. The non-linear model (ELM) was superior to the linear method (PLSR). ELM demonstrated satisfactory accuracy based on the feature wavelet coefficients of CWT with the mexh wavelet basis function at a decomposition scale of 1 (CWT(mexh_1)), with $R^2$, RMSE, and RPD values of 0.946, 1.408%, and 3.759 in the validation dataset, respectively. Overall, the CWT(mexh_1)-CARS-ELM systematic modeling method was feasible and reliable for estimating the water content of sandy clay loam.

# INTRODUCTION

Soil water content affects the growth, development, and physiological processes of plants (*Crusiol et al., 2023*). Accurately monitoring soil water content can provide better guidance for agricultural production, improve water resource utilization, and contribute to sustainable agricultural development. There are many traditional methods for measuring soil water content, such as the oven-drying method, neutron method, $\gamma$ ray method, and dielectric method. However, these methods have different drawbacks in practical measurements, such as damaging the inherent properties of the soil or requiring a perennial measurement cycle; some radiation methods can even be hazardous to human health (*Zhao & Wang, 2002*). Hyperspectral technology has therefore attracted the interest of numerous researchers because of its real-time, high efficiency, and non-destructive properties (*Zhang et al., 2023a*).

The hyperspectral data obtained from spectral devices are abundant and span multiple bands. However, there is often a lot of noise interference, which needs to be eliminated through pre-processing (*Yuan et al., 2019*). The results of previous studies indicate that methods such as Savitzky-Golay (SG) smoothing, moving average, and scattering correction are widely applied in the pre-processing of spectral data (*Chu, Yuan & Lu, 2004*). However, achieving satisfactory results in handling white noise with these methods remains a challenge (*Cai & Ding, 2017*; *Zhang, Li & Pan, 2017*). Wavelet transform can improve the sensitivity between spectra and target parameters compared to traditional mathematical algorithms. A previous study used discrete wavelet transform (DWT) with a db4 mother wavelet to handle noisy data in the spectra of different soil organic matter content (*Meng et al., 2021*). The best denoising effect was achieved, resulting in the optimal predictive model accuracy, when DWT was combined with a 0.6-order derivative to process the spectra. The spectral data of soils with different heavy metal concentrations were pre-processed by SG smoothing, and spectral features were enhanced with continuous wavelet transform (CWT) with the gaus4 wavelet basis function. CWT decomposition scales of 1 to 5 had a more significant effect on spectral noise reduction and peak features than CWT decomposition scales of 6 to 8 (*Zhang et al., 2022*). In terms of spectral application in soil water content inversion, a combination of db10 wavelet packet transform (WPT) and harmonic analysis was used to pre-process the spectral data, successfully smoothing and compressing the data and achieving significantly better soil water content inversion accuracy than other approaches (*Jiang et al., 2017*). Therefore, further exploration is needed in soil water content inversion to select wavelet basis functions and determine decomposition scales when applying CWT.

Spectral indices of hyperspectral data were established by extracting partial bands for inversion objective parameters (*Ren et al., 2022*). Spectral indices were used for soil water content estimation; higher accuracy was achieved under lower moisture conditions, but spectral index saturation rose once the soil water content became too high (*Yan et*

*al., 2023*). Some researchers have incorporated full-band hyperspectral data into model development and achieved satisfactory predictive accuracy, but this also increases the number of parameters to be estimated in the model (*Liu et al., 2023*). The full-band information of soil spectral reflectance may be the comprehensive result of various factors, such as soil water, particle size, organic matter, and salinity, hindering the spectral retrieval of soil water content. Researchers have already analyzed the characteristic spectra of soil water content, salinity, organic matter, et al., using statistical methods or intelligent algorithms (*Tang et al., 2021*; *Xia et al., 2021*; *Jia et al., 2022a*; *Seema Ghosh et al., 2022*), such as correlation analysis, stepwise regression, competitive adaptive reweighted sampling (CARS), successive projections algorithm (SPA), variable importance in projection (VIP), uninformative variable elimination (UVE), and random frog algorithm (Rfrog). Overfitting can be avoided in model building by screening variables using UVE. However, the number of screened variables is still too large with UVE, and secondary extraction is needed. CARS can effectively screen out important wavelength variables (*Ye, Wang & Min, 2008*; *Yu et al., 2016b*). The SPA algorithm can effectively avoid overlap of spectral information, but because of the non-collinear effective spectral information of soil, the extracted feature band information cannot all be expressed by the SPA algorithm, resulting in the loss of some information (*Yu et al., 2016b*) and affecting the prediction accuracy of the model. Moreover, the predictive ability and stability of SPA are not as strong as the predictive ability and stability of CARS (*Guo et al., 2023*). CARS has been widely used in spectral analyses to predict soil organic matter, soil water content, plant nitrogen content, and leaf water content (*Cai & Ding, 2018*; *Sun et al., 2021*; *Xing et al., 2021*). CARS can optimize effective wavelength variables, greatly reduce the dimension of hyperspectral variables and the degree of computational complexity, and improve the prediction ability of the model (*Yu et al., 2016c*). It is important to select an efficient and stable feature extraction method to reduce data dimension and improve model efficiency.

In the estimation model of soil composition-related indicators, soil organic matter content was predicted using partial least squares regression (PLSR; $R^2 = 0.76$) (*Sun et al., 2022*). The random forest (RF) and extremely randomized trees (ERT) models had excellent advantages in predict soil salinity indicators (*Jia et al., 2022b*). The inversion of soil water using Hapke photometric models contributed to soil surface water monitoring (*Zhang et al., 2020*). Combining the first derivative and normalized difference spectral index achieved a good fit for soil water prediction ($R^2 = 0.977$) (*Yan et al., 2023*). Combining the entire subset screening with machine learning methods achieved a good inversion accuracy of soil water content and improved the robustness ($R^2 = 0.750$) (*Tan et al., 2020*). In terms of soil water content prediction, the linear model showed good predictive ability ($R^2 = 0.65$) for soil water content when the soil sample water content was lower than the field capacity, and the non-linear model RF further improved the prediction accuracy ($R^2 = 0.75$) (*Liu et al., 2023*). However, previous research results have indicated a negative correlation between soil spectral reflectance and soil water content when soil water content falls below field capacity conditions (*Ma & Fan, 2020*). Additionally, when the soil water content is more than field capacity, there may be specular reflection characteristics, and soil spectral reflectance is positively correlated with water content (*Bablet et al., 2018*), with the threshold of this trend

around field capacity (*Liu et al., 2002b*). The relationship between soil spectral reflectance and water content is non-linear, and further exploration is needed of suitable prediction model building methods when there is a wide range of soil water content levels (including cases where some samples have moisture content higher than the field capacity and other samples have lower moisture content than the field capacity).

In this paper, the hyperspectral data of ground and sieved soil samples were processed by CWT, and CARS feature extraction and prediction models were established using PLSR and extreme learning machine (ELM). The main objectives of this study were: (1) to reduce noise in hyperspectral data based on CWT (with different wavelet basis functions and decomposition levels), and enhance the correlation between soil water content and wavelet coefficients; (2) to extract the spectral wavelengths corresponding to the feature wavelet coefficients using CARS, reduce redundant information, and improve modeling accuracy; (3) to establish the soil water inversion models based on the full-band wavelet coefficients and the selected feature band wavelet coefficients, respectively, and evaluate the accuracy of the models to identify the best wavelet basis function and decomposition level.

## MATERIALS AND METHODS

### Preparation of soil samples

The soil used in this study was collected from the topsoil layer (0–20 cm) of an uncropped experimental field at the Kunming University of Science and Technology, Kunming, Yunnan, China. According to the international soil classification system, the soil was classified as sandy clay loam with 20.03% clay, 62.32% silt, and 17.65% sand. The field-collected topsoil samples were subjected to drying, grinding, mixing, impurity removal, and sieving (two mm) to minimize the influence of soil particle size on the data collection. The soil samples with an extensive range of water content levels were prepared in the laboratory. The samples were prepared by placing the soil (porosity: 61.65%, bulk density: 1.01 g/cm$^3$, organic matter: 0.75%) in a disk with an inner diameter of 16 cm, a height of 1.7 cm, and fine holes at the bottom. The surface of the soil was leveled, and then the disk was placed in water that was approximately one cm deep. The soil sample was subjected to water absorption through the circular hole at the bottom of the disc until it reached saturation. The disc was then removed from the water and placed on dry soil for gravity drainage. After various time intervals, 139 samples were obtained with water content ranging from 13.48% to 47.88%. This treatment method effectively addresses the issue of uneven surface caused by adding water from the soil surface, which affects soil reflectivity (*Wang et al., 2023c*).

### Data acquisition

The hyperspectral reflectance information of the soil samples was measured using the SR-2500 portable spectroradiometer with an 8° field of view optical fiber (Spectral Evolution, Inc., Lawrence, MA, USA). The spectral measurement range of the equipment was 350–2,500 nm, with a spectral resolution of 3.5 nm and a sampling interval of 1.5 nm between 350–1,000 nm, and a spectral resolution of 22 nm and a sampling interval of 6 nm between

1,000–2,500 nm. The instrument interpolated the measurement data into 1 nm intervals. Sunny weather conditions were chosen to obtain stable light sources, and measurements were carried out from 10:00 to 14:00 local time to ensure an appropriate solar altitude. The instrument was calibrated using a standard whiteboard prior to measurement, and whiteboard calibration was performed every 10 min during operation. Sampling was done by placed the fiber optic 15 cm above the soil sample, vertically, with the fiber optic's field of view covering no more than the disk range. Each soil sample was scanned ten times and the average spectral data value was used as the spectral reflectance of the sample. After the spectral data was obtained, the soil water content of soil samples was measured using the oven-drying method in the laboratory (*Cheng et al., 2021*). The field capacity of the soil was measured using the Wilcox method (mass moisture content 31.63%).

## Sample division

The sample data in this study was divided into training and validation datasets in a ratio of 2:1 based on the joint X–Y distance (SPXY) method. SPXY is an extension of the Kennard-Stone algorithm and is a common method of sample division. SPXY simultaneously considers the spectral space (x) and the feature space (y) when calculating the distance between samples, and has demonstrated good performance in establishing a quantitative inversion model (*Galvao et al., 2005*). The equations are shown as follows:

(1) The Euclidean distance of the spectral values (x) is calculated using Eq. (1)

$$d_x(p,q) = \sqrt{\sum_{j=1}^{I}[x_p(j) - x_q(j)]^2}; p,q \in [1,N] \tag{1}$$

(2) The Euclidean distance of the dependent variable (y) is calculated using Eq. (2):

$$d_y(p,q) = \sqrt{(y_p - y_q)^2} = |y_p - y_q|; p,q \in [1,N] \tag{2}$$

(3) The normalized distance is calculated using Eq. (3):

$$d_{xy}(p,q) = \frac{d_x(p,q)}{\max_{p,q \in [1,N]} d_x(p,q)} + \frac{d_y(p,q)}{\max_{p,q \in [1,N]} d_y(p,q)}; p,q \in [1,N] \tag{3}$$

where $x_p$ and $x_q$ are the spectral reflectance values of the samples at the $j$ th wavelength, respectively; $I$ represents the number of spectral wavelengths; and $N$ represents the number of samples.

## Data processing

In the measured spectral curve (350–2,500 nm), there was a strong water absorption band around 1,400 nm, and the spectral absorption characteristics of water were gradually enhanced as soil water content increased, with reflectance at wavelengths greater than 1,400 nm showing poor stability (*Bogrekci & Lee, 2006*; *Ogen et al., 2019*). To reduce the interference of atmospheric moisture on spectral information (*Yuan et al., 2019*), a spectral range of 350–1,349 nm was selected for this study (*Zhang et al., 2020*). The continuous wavelet transform (CWT) method, which is a type of linear transformation and is commonly used in digital signal processing, was used for data pre-processing in
this study. The main feature of the CWT method is that transformation can highlight certain signal characteristics and perform a localized analysis of time (space) frequency (*Li et al., 2021b*). The signal is then refined at multiple scales through scaling and translation operations, resulting in higher time resolution at high frequencies and higher frequency resolution at low frequencies. Thus, any signal details can be focused on, and better signal mutations can be observed (*Zhang et al., 2016*). The wavelet transform involves scale factor "a" and translation factor "b", where the scale factor "a" controls the scaling deformation of the function, and the translation variable "b" can change the position of the function. The transformation process is implemented using Eqs. (4) and (5):

$$Wf(a,b) \leq f; \psi_{a,b} \geq \int_{-\infty}^{+\infty} f(\lambda)\psi_{a,b}(\lambda)d\lambda \tag{4}$$

$$\psi_{a,b}(\lambda) = \frac{1}{\sqrt{a}}\psi\left(\frac{\lambda-b}{a}\right) \tag{5}$$

where $\lambda$ is the selected soil hyperspectral wavelength ($350\sim1,349$ nm); $f(\lambda)$ is the original hyperspectral reflectance data; $Wf(a,b)$ is the wavelet coefficient, which consists of an n $\times$m matrix composed of wavelet coefficients in two dimensions, the decomposition level $i$ ($i = 1, 2,\ldots$, n) and the band $j$ ($j = 1, 2,\ldots$, m), respectively; and $\psi_{a,b}(\lambda)$ is the wavelet basis function.

In this study, CWT was used to preprocess the hyperspectral information, and the wavelet coefficients of multiple wavelet basis functions were analyzed for correlation with soil water content. The wavelet basis functions with high correlation, gaus2, and mexh, were selected to carry out the subsequent studies (*Tan et al., 2021*; *Wang et al., 2023a*). The gaus2 wavelet basis function is the differential form of the Gaussian density function, which is a non-orthogonal and non-biorthogonal wavelet without a scaling function. After the CWT with gaus2 wavelet basis function, the contrast of the image correlation peak was enhanced, suppressing interference and improving image processing (*Liu, Wang & Huang, 2022*). The mexh wavelet basis function has good localization in both the time and frequency domains and is capable of detecting abrupt changes in the signal and highlighting the characteristic points containing information (*Zhang et al., 2023b*).

## Feature variable extraction

The competitive adaptive reweighted sampling (CARS) algorithm was used in this study for feature variable extraction of wavelet coefficients. CARS is a feature variable selection method that combines Monte Carlo sampling with partial least squares regression (PLSR), imitating the "survival of the fittest" principle in Darwinian theory. In the CARS algorithm, each iteration involves adaptive reweighted sampling (ARS) and exponentially decreasing function (EDP) to retain the samples with higher absolute value weight of regression coefficients in the PLSR model as a new subset, while eliminating points with smaller weights. A new PLSR model is then built based on this new subset. After multiple calculations are performed, the wavelength subset corresponding to the wavelet coefficients, which minimizes the root mean square error of cross-validation (RMSECV) of the PLSR model, is selected as the feature variable (*Nirere et al., 2023*).

## Modeling method

This study used the linear model (PLSR) and non-linear model (extreme learning machine, ELM) for soil water content prediction model building.

PLSR is a commonly used approach in chemometrics and quantitative spectroscopic analysis and combines the advantages of principal component analysis, classical correlation analysis, and linear regression analysis. PLSR addresses multicollinearity issues among independent variables and when there is a smaller number of samples than variables, achieving a fast model training speed (*Asante et al., 2021*). The number of latent variables (LVs) is a primary parameter of the PLSR model, impacting model results. In this study, the number of LVs was determined using a ten-fold cross-validation method (*Zhang et al., 2019*). The expression of the PLSR model is as follows:

$$X = TP' + E, Y = UQ' + F, U = \beta T \tag{6}$$

where $X$ represents the independent variable matrix; $Y$ represents the response matrix; $T$ represents the score matrix for $X$; $U$ represents the transposed score matrix for $Y$; $P$ represents the loading matrix of $X$; $Q$ represents the transposed loading matrix of $Y$; $E$ represents the residual matrix for $X$; $F$ represents the transposed residual matrix for $Y$; and $\beta$ represents the regression coefficient matrix.

ELM is a feed-forward neural network that has been proven to have learning efficiency and strong generalization ability due to the characteristics of the weights and biases for the input layer and hidden layer, which are randomly generated within a given range. The primary purpose of training is to solve the output weights $\beta$. Compared to the traditional single-hidden-layer feed-forward neural network (SLFN), even with randomly generated hidden nodes, ELM still maintains the universal approximation capability of SLFN. The connection weights between the input and hidden layers of ELM are randomly generated without manual adjustment during the training process (*Huang, Zhu & Siew, 2006*). As a new learning framework for single-hidden-layer feed-forward neural networks, ELM integrates advantages such as regression, classification, clustering, and compression in machine learning. The schematic structure of ELM is shown in Fig. 1.

Given a set of training samples $N$, where $x_i \in R^n$ represents the input values and $t_i \in R^n$ represents the target output values, when the number of hidden layer nodes is $L$, the output expression of ELM is as follows:

$$f_L(x) = \sum_{i=1}^{L} \beta_i g_i(x) = \sum_{i=1}^{L} \beta_i g_i(w_i \times x_i + b_i), j = 1, \ldots, N \tag{7}$$

where $f_L(x)$ represents the network output values, $\beta_i$ is the weight vector between the ith hidden layer neurons and the output layer, $w_i$ is the weight vector between the input and output layers, $b_i$ is the bias vector, and $g_i(w_i \times x_i + b_i)$ is the activation function of the $i$ th hidden nodes.

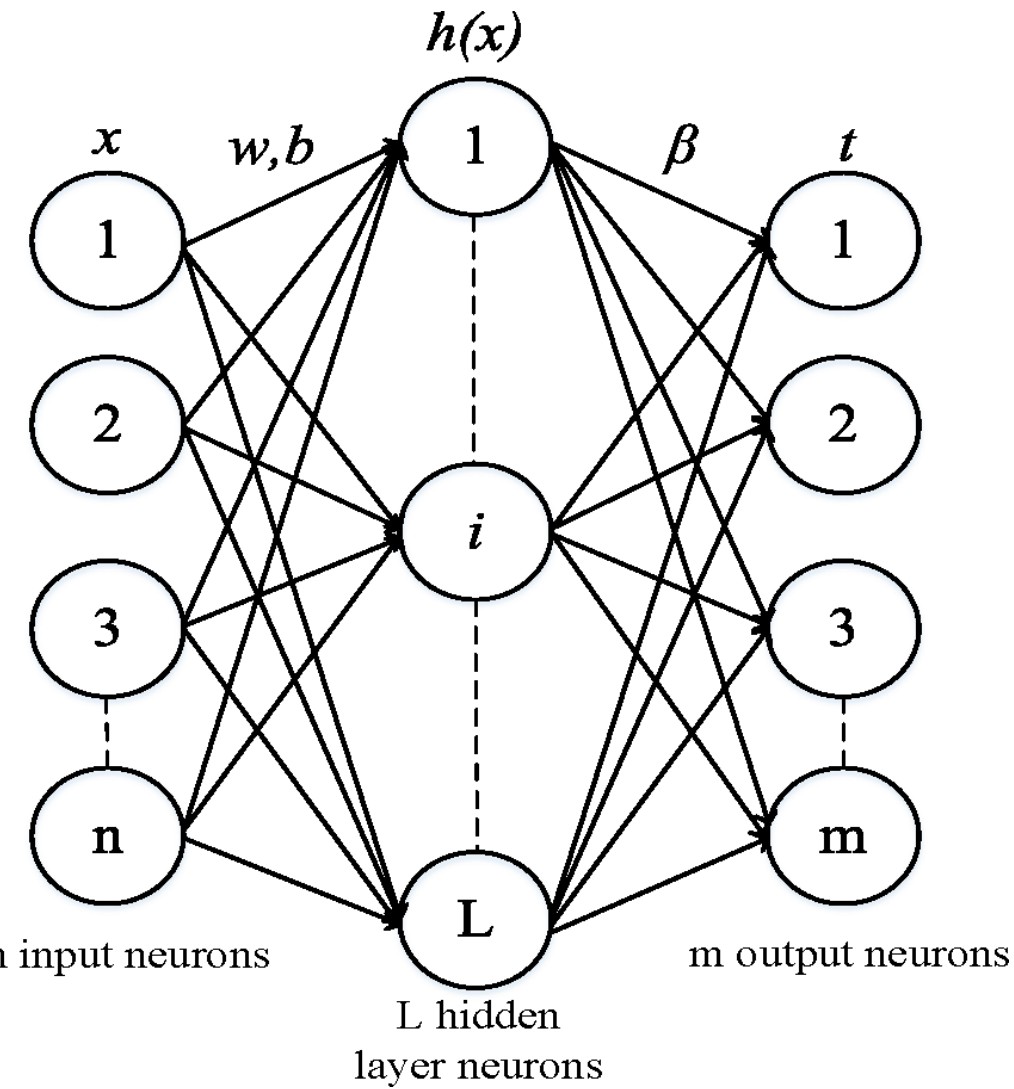

**Figure 1** **Extreme learning machine structural diagram.** $\omega$ is weight vector, $b$ is the bias vector, and $\beta$ is the weight vector.

When the activation function $g_i(w_i \times x_i + b_i)$ can approximate the N samples with zero error, it can be shown in the following equations:

$$\sum_{j=1}^{L} \|f_L(x) - t_i\| = 0 \tag{8}$$

$$\sum_{i=1}^{L} \beta_i g_i(w_i \cdot x_i + b_i) = t_i \tag{9}$$

where $t_i$ is the output target value.

Equations (8) and (9) can be abbreviated to Eq. (10), as follows:

$$T = H\beta \tag{10}$$

where H is calculated using Eq. (11), and $\beta$ and $T$ are calculated using Eq. (12), as follows:

$$H = \begin{bmatrix} g(w_1 \times x_1 + b_1) & \cdots & g(w_L \times x_1 + b_L) \\ \vdots & \cdots & \vdots \\ g(w_1 \times x_N + b_1) & \cdots & g(w_L \times x_N + b_L) \end{bmatrix}_{N \times L} \tag{11}$$

$$\beta = \begin{bmatrix} \beta_1^T \\ \vdots \\ \beta_L^T \end{bmatrix}_{L \times m} \quad T = \begin{bmatrix} t_1^T \\ \vdots \\ t_N^T \end{bmatrix}_{N \times m} \tag{12}$$

where $m$ represents the output value, $H$ represents the output matrix of the hidden layer, and $T$ represents the output target matrix.

Because the $H$ matrix is invertible, the weight vector $\beta$ between the hidden layer and the output layer can be calculated by Eq. (13):

$$\hat{\beta} = H^+ T \tag{13}$$

where $H^+$ represents the generalized inverse of $H$.

In summary, the main steps of the ELM algorithm are as follows:

(1) Determine the number of hidden layer neurons $L$ and randomly assign the input weight matrix $w$ and hidden layer bias matrix $d$.

(2) Choose an appropriate activation function $g_i(w_i \times x_i + b_i)$ for the hidden layer and further calculate the hidden layer output matrix $H$.

(3) Calculate the weight vector $\hat{\beta}$ of the output layer.

In this study, the default 'sigmoid' was set as the activation function. The number of hidden layer neurons (HLNs) gradually increased from 5 to 100, with a step size of 1. Each ELM model was run 500 times to determine the optimal number of HLNs based on the best training results.

## Model performance evaluation

The determination coefficients ($R^2$), root mean square error (RMSE), and relative prediction deviation (RPD) were used to evaluate the performance of models (*Shi et al., 2014*) and were calculated using the following equations:

$$R^2 = \left[ \frac{\sum_{i=1}^{n}(y_i' - \overline{y'})(y_i - \overline{y})}{\sqrt{\sum_{i=1}^{n}(y_i' - \overline{y'})^2}\sqrt{\sum_{i=1}^{n}(y_i - \overline{y})^2}} \right]^2 \tag{14}$$

$$RMSE = \sqrt{\frac{\sum_{i=1}^{n}(y_i' - y_i)^2}{n}} \tag{15}$$

$$RPD = \frac{STD}{RMSE} \tag{16}$$

where $y_i'$ represents the ith simulated value, $\overline{y'}$ represents the average of the simulated value, $y_i$ represents the ith measured value, $\bar{y}_i$ represents the average value of the measured value,

**Table 1  Descriptive statistical characteristics of the soil sample water content.**

| Dataset | Sample size | Sample water content /% | | | | CV/% |
|---|---|---|---|---|---|---|
| | | Maximum | Minimum | Mean | STD | |
| Population dataset | 139 | 47.88 | 13.48 | 28.52 | 7.69 | 26.96 |
| Training dataset | 93 | 47.88 | 13.48 | 27.33 | 8.38 | 30.66 |
| Validation dataset | 46 | 40.22 | 16.17 | 30.94 | 5.39 | 17.42 |

Notes.
STD is the standard deviation; CV is the coefficient of variation of the dataset.

$n$ represents the number of data points, and $STD$ represents the standard deviation of the predicted value. Larger $R^2$ and $RPD$ values and smaller $RMSE$ values indicate better model performance.

## RESULTS

### Soil water content and hyperspectral reflectance characteristics

In this study, a total of 139 soil samples were prepared, including 93 training samples and 46 verification samples. The prepared soil samples had a relatively wide range of water content, ranging from 13.48% to 47.88% (Table 1), with an average value of 28.52%, a standard deviation of 7.69%, and a coefficient of variation of 26.96%. The water content of the training dataset had a range of 13.48–47.88%, and the water content of the validation dataset had a range of 16.17–40.22%. The data range of the validation dataset was covered by the training dataset, meeting the requirements of model establishment and guaranteeing the accuracy of the models.

The hyperspectral reflectance curves of soil samples with different water content percentages are shown in Fig. 2. The spectral reflectance value increased rapidly as wavelength increased in the range of 500–750 nm. The reflectance curves of each sample exhibited a reflection peak around 770 nm, with tending to stabilize after 880 nm. The samples were divided into five gradients by water content, and the reflectance data within each gradient range were plotted in the shadow region. The average reflectance of each gradient was highlighted in the form of a dark curve, and the soil water content corresponding to the average value was identified. The soil sample with a water content of 31.58% (close to the field capacity) was considered the inflection point: when the soil water content was less than this value, the soil reflectance decreased as water content increased; conversely, when the soil water content exceeded 31.58%, the soil reflectance increased as water content increased. Therefore, the soil reflectance curve first decreased and then increased as soil water content increased, with an inflection point when the soil water content was near the field capacity.

### Correlation analysis between CWT wavelet coefficient and soil water content

The original hyperspectral reflectance was processed by CWT with mexh and gaus2 wavelet basis functions to achieve wavelet coefficients at a decomposition scale range of 1–10. The

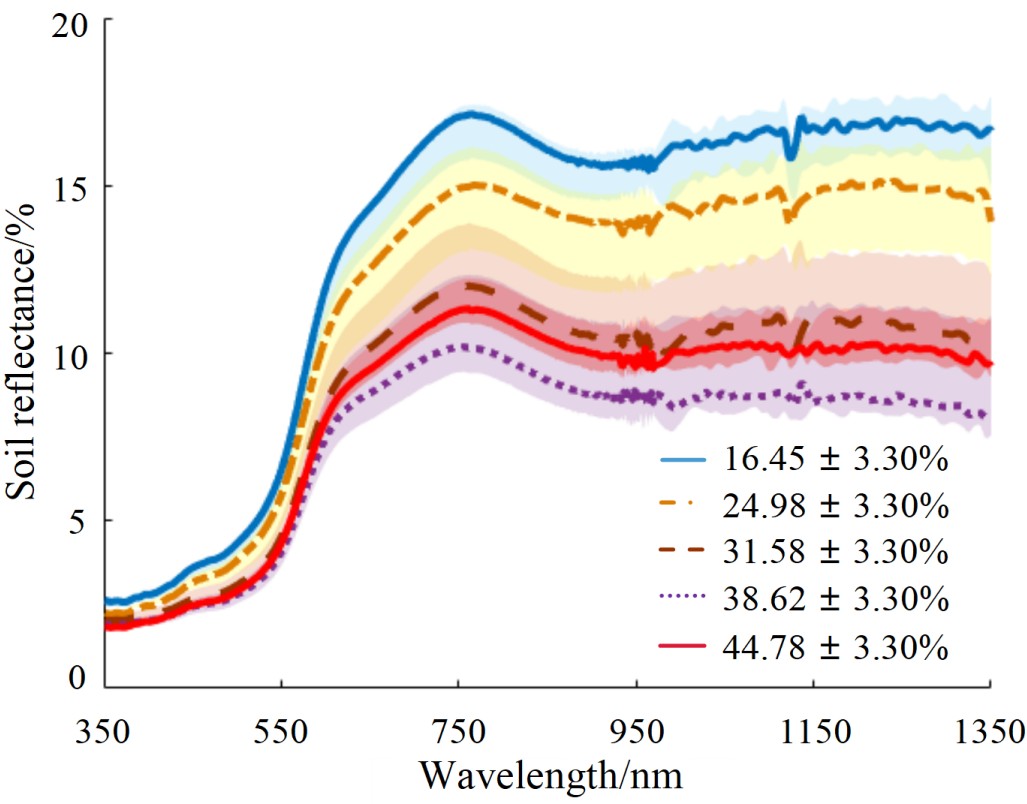

**Figure 2** Average reflectance curves for five soil water content gradients.

correlation spectrum between the wavelet coefficients and soil water content is illustrated in Fig. 3.

According to Fig. 3A, at the same wavelength, the correlation between wavelet coefficients and the soil water content gradually decreased as the decomposition scale increased from 1 to 10. The mexh wavelet basis function correlated better at decomposition scales of 1–5. At decomposition scales of 4 and 5, when the wavelengths were in the range of 850–1,336 nm, the absolute value of the correlation coefficients ($|r|$) was in the range of 0.50–0.60. At decomposition scales of 1 to 3, the wavelengths were around 1,104–1,118 nm and 1,157–1,336 nm, and the $|r|$ ranged from 0.60 to 0.69. When the decomposition scale was 1, the maximum value of $|r|$ was approximately 0.68, at a wavelength of around 1,331 nm.

Figure 3B shows the correlation between soil water content and wavelet coefficients of the gaus2 wavelet basis function. Similar to the results of the mexh wavelet basis function (Fig. 3A), the decomposition results of gaus2 also showed that, at the same wavelength, the correlation coefficients decreased as the decomposition scale increased. The $|r|$ was 0.50–0.60 within the gaus2 wavelet basis function at decomposition scales of 1–5, mainly distributed between wavelengths of 1,006–1,115 nm and 1,138–1,335 nm. When the decomposition scale was 1, the maximum value of $|r|$ was 0.59, at a wavelength of around 1,333 nm.

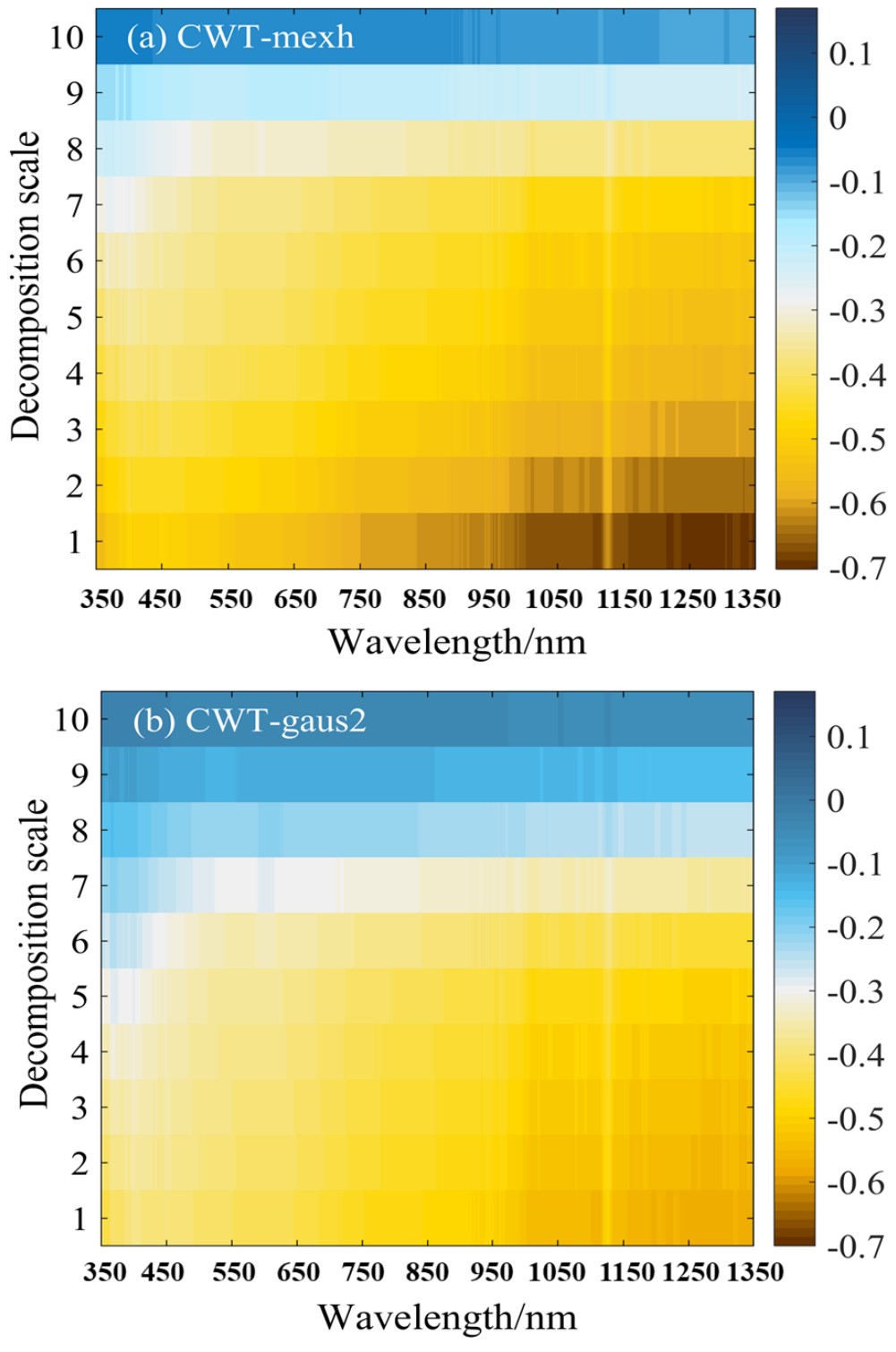

**Figure 3** (A–B) Correlation between soil water content and the wavelet coefficients of two different wavelet basis functions (mexh, gaus2) at 10 decomposition scales.

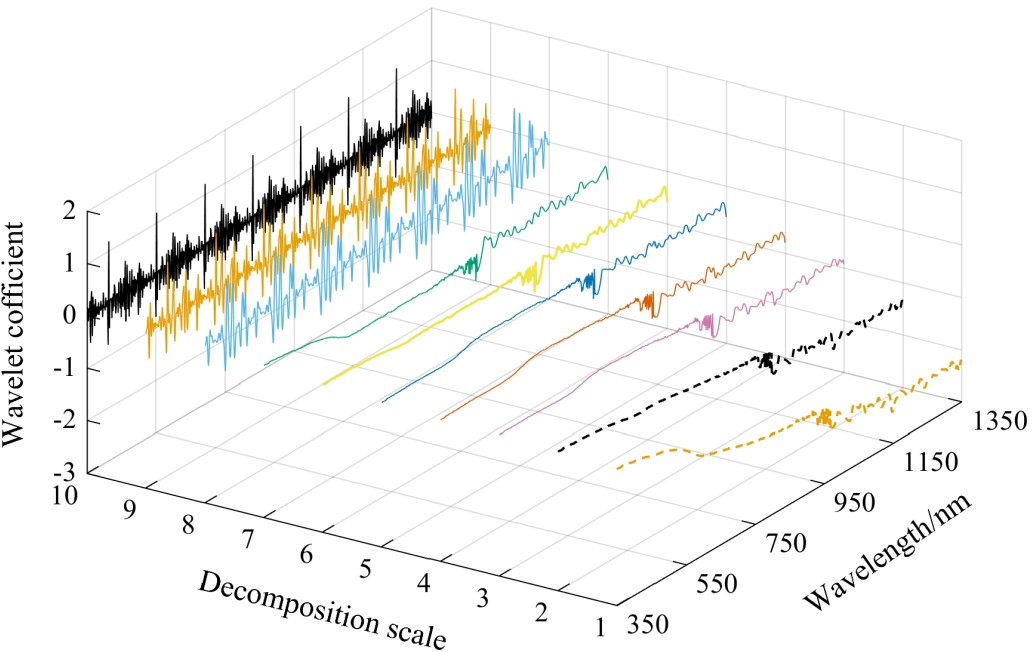

**Figure 4** The mexh wavelet coefficient curves of a soil sample at 10 decomposition scales (soil water content is 31.62%).

Therefore, based on the results of the correlation analysis, the mexh and gaus2 wavelet coefficients in decomposition scales of 1–5 were selected for subsequent studies to obtain a soil water content prediction model with strong predictive capability.

The wavelet coefficient curves of the spectral curve with a soil water content of 31.62% were plotted as an example using the mexh wavelet basis function and CWT to decompose soil spectral reflectance data to decompression scales of 1–10 (Fig. 4). As shown in Fig. 4, the wavelet coefficient curves gradually curved down, and the absolute values of wavelet coefficients increased as the wavelength increased at decomposition scales of 1–7. The curves at decomposition scales of 8, 9, and 10 changed dramatically and showed over-decomposition, resulting in both a weakened correlation between soil water content and wavelet coefficients at these three scales and lower correlation coefficients (Fig. 3).

## Extracted feature wavelet coefficients

The feature wavelet coefficients were selected using the CARS method. Figure 5 shows the feature extraction process. Figure 5A shows that the variable extraction trend went from fast to slow, representing the two stages of "rough selection" and "refinement selection" in CARS. Using the mexh wavelet coefficients at a decomposition scale of 1 as an example, when the number of sampling runs reached 33, RMSECV reached its minimum value (1.3; Fig. 5B), suggesting that a significant number of irrelevant wavelengths were eliminated during the sampling process from runs 1 to 33. However, with more than 33 sampling runs, the RMSECV gradually increased, indicating some wavelet coefficients related to soil water content were excluded. The colored lines in Fig. 5C represent the changing trend for
each wavelength of regression coefficients with the sampling operations. Each sampling operation records a subset of wavelengths, and the position of the blue asterisk represents the optimal subset of wavelet coefficients. Based on the result shown in Fig. 5, the number of feature wavelet coefficients is 17. The corresponding bands can be further extracted to obtain the feature wavelet coefficients of the mexh wavelet at the decomposition scale of 1.

The wavelength distribution corresponding to the selected feature wavelet coefficients of the mexh and gaus2 wavelet basis functions at different decomposition scales is displayed in Fig. 6. Figure 6A shows that the wavelengths corresponding to the feature wavelet coefficients for the mexh wavelet of a decomposition scale of 1 were relatively concentrated, mainly distributed in the range of 950–1,340 nm. When the decomposition scale was 2, the highest quantity of feature wavelet coefficients was extracted, primarily concentrated in the ranges of 590–610, 930–1,035, and 1,140–1,230 nm. As shown in Fig. 6B, for the gaus2 wavelet basis function, the largest number of feature wavelet coefficients was obtained when the decomposition scale was 1, with these feature wavelet coefficients mainly concentrated in the ranges of 360–660, 950–1,190, and 1,330–1,340 nm. Overall, the results of the two wavelet basis functions showed that the distribution of feature wavelet coefficients between 700 and 900 nm was sparse or even non-existent, and the feature wavelet coefficients were most dense between 930 and 1,340 nm. All of these results indicate that the feature wavelet coefficients were predominantly concentrated in the near-infrared region (>900 nm).

## Model establishment based on wavelet coefficients and evaluation

The full-band wavelet coefficients and the feature wavelet coefficients selected by CARS were applied as input variables to build the PLSR and ELM models.

Prediction models were established using the full-band wavelet coefficients obtained from CWT with mexh and gaus2 wavelet basis functions (Table 2). The validation dataset results showed that as the wavelet decomposition scale increased, the $R^2$ and RPD values of the PLSR models based on the full-band wavelet coefficients decomposed by mexh and gaus2 gradually decreased, and the accuracy of the model gradually decreased. A better model result was achieved using the wavelet coefficients of the mexh wavelet basis function at a decomposition scale of 1 ($R^2 = 0.867$, RPD = 2.130). Similarly, the accuracy of the non-linear ELM model also decreased as the wavelet decomposition scale increased from 1 to 5. The highest model accuracy, with an $R^2$ of 0.925 and RPD of 2.947, was obtained using the full-band wavelet coefficients of the mexh wavelet basis function at a decomposition scale of 1.

The results of the PLSR and ELM models established based on the extracted feature band wavelet coefficients are shown in Table 3. The PLSR models were established based on the feature wavelet coefficients of two wavelet basis functions; the $R^2$ and RPD values decreased as the decomposition scale increased. A better result was achieved by modeling the feature wavelet coefficients of the mexh wavelet basis function at a decomposition scale of 1, reaching an $R^2$ of 0.866 and RPD of 2.209. Based on the same dataset, the overall trend of the $R^2$ and RPD values of ELM models also decreased as the decomposition scale increased. A better result was obtained by modeling the feature wavelet coefficients of the mexh wavelet basis function at a decomposition scale of 1 ($R^2 = 0.946$, RPD = 3.759).

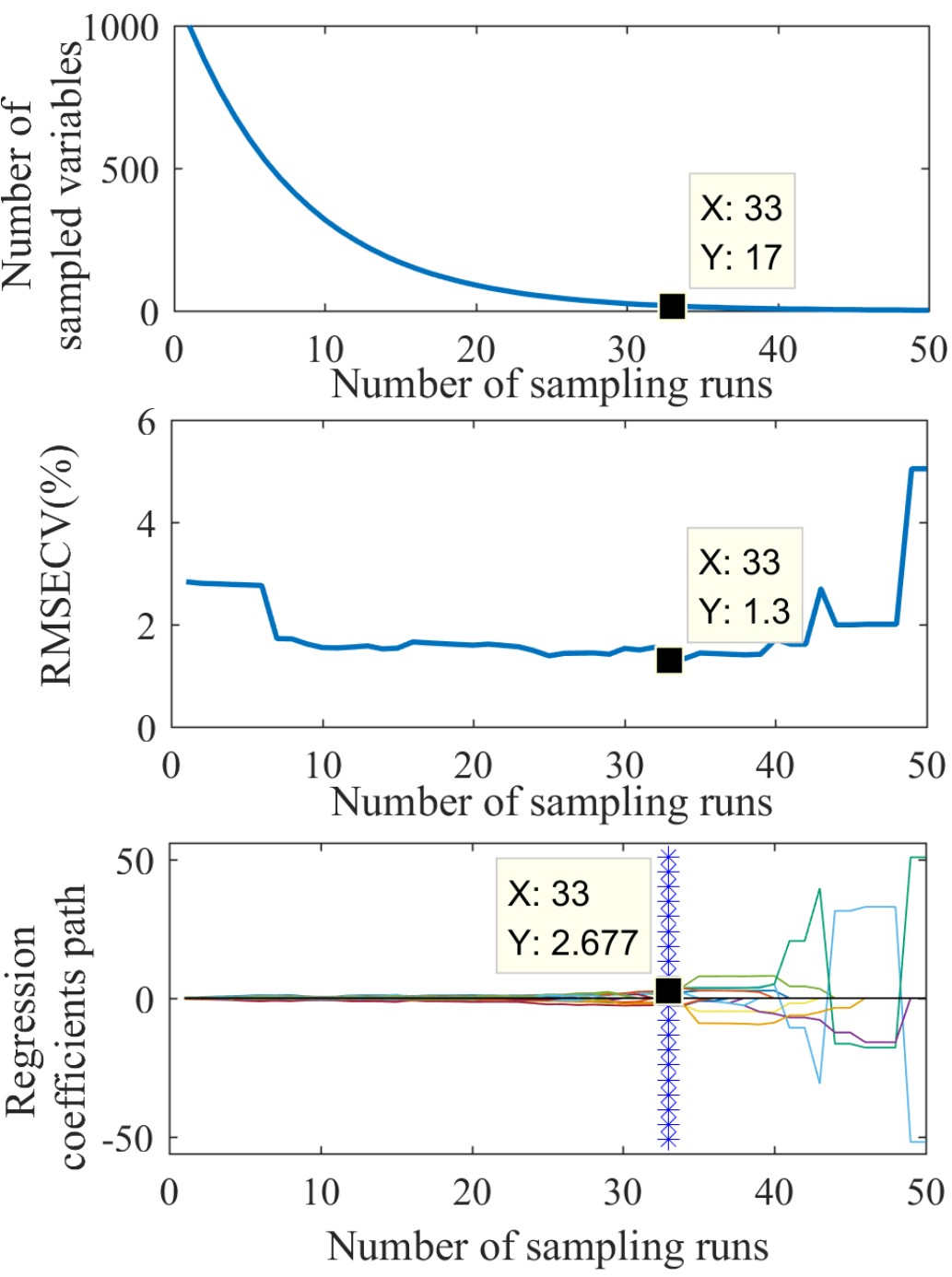

**Figure 5** The process of feature extraction by CARS.

Based on the feature wavelet coefficients, both the PLSR and ELM methods achieved superior results under the first decomposition scale of the mexh wavelet basis function (Table 3). The scatter plots of the measured and predicted values of the PLSR and ELM models are shown in Fig. 7. The slopes of the fitted lines for the PLSR and ELM models in

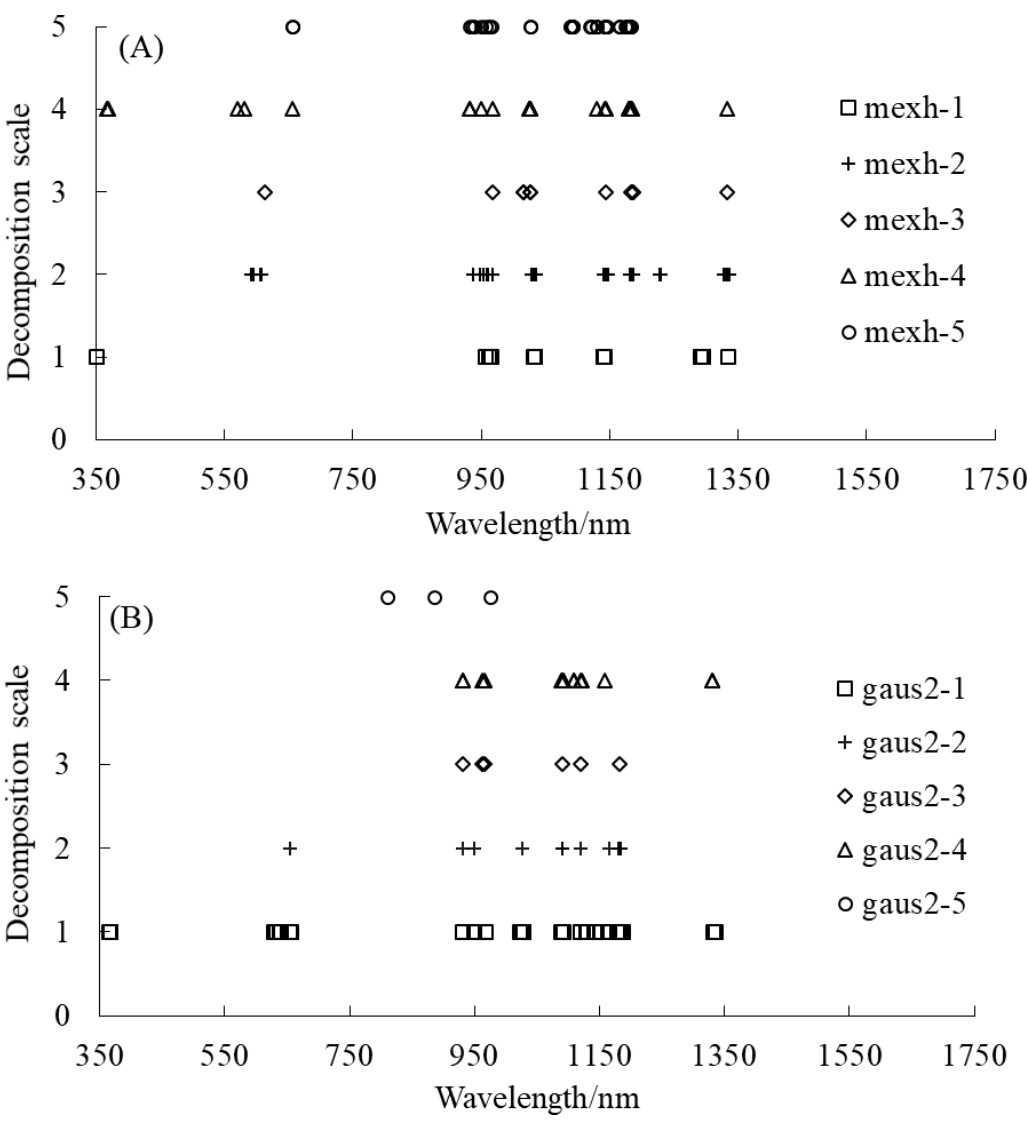

**Figure 6** **The wavelength distribution corresponding to the selected feature wavelet coefficients.** (A) mexh, (B) gaus2.

the validation dataset (Figs. 7B and 7D) were 1.047 and 1.029, respectively, indicating that neither model was significantly overestimated or underestimated and that both the PLSR and ELM models are reliable and accurate. The slope of the fitted line for the ELM model was closer to 1 than that of the PLSR model. However, when the soil water content was higher than the field capacity, the prediction errors of the PLSR model were larger than those of the ELM model, PLSR model had a more dispersed scatter plot (Figs. 7B and 7D, part of the dotted circle). Therefore, the optimal modeling method was the ELM model using the feature wavelet coefficients extracted by the CARS method from the CWT with the mexh wavelet basis function at a decomposition scale of 1.

**Table 2 Modeling results based on the full-band wavelet coefficients.**

| Modeling method | Wavelet basis function | Scale | Training dataset | | Validation dataset | | | LVs/HLNs |
|---|---|---|---|---|---|---|---|---|
| | | | $R^2$ | RMSE(%) | $R^2$ | RMSE(%) | RPD | |
| PLSR | mexh | 1 | 0.909 | 2.509 | 0.867 | 2.718 | 2.130 | 8 |
| | | 2 | 0.717 | 4.423 | 0.655 | 3.419 | 1.600 | 2 |
| | | 3 | 0.707 | 4.519 | 0.558 | 4.169 | 1.249 | 2 |
| | | 4 | 0.677 | 4.747 | 0.477 | 4.477 | 1.275 | 2 |
| | | 5 | 0.645 | 4.928 | 0.387 | 5.071 | 1.221 | 2 |
| | gaus2 | 1 | 0.680 | 4.652 | 0.533 | 4.667 | 1.312 | 2 |
| | | 2 | 0.667 | 4.792 | 0.389 | 4.959 | 1.235 | 3 |
| | | 3 | 0.638 | 4.930 | 0.342 | 5.461 | 1.180 | 2 |
| | | 4 | 0.588 | 5.203 | 0.231 | 5.925 | 1.132 | 2 |
| | | 5 | 0.506 | 5.634 | 0.169 | 6.605 | 1.041 | 2 |
| ELM | mexh | 1 | 0.941 | 2.191 | 0.925 | 1.728 | 2.947 | 16 |
| | | 2 | 0.934 | 2.454 | 0.911 | 1.515 | 2.679 | 16 |
| | | 3 | 0.922 | 2.696 | 0.905 | 1.571 | 2.468 | 20 |
| | | 4 | 0.919 | 2.657 | 0.878 | 2.137 | 2.608 | 20 |
| | | 5 | 0.881 | 3.327 | 0.872 | 3.116 | 2.505 | 18 |
| | gaus2 | 1 | 0.932 | 2.556 | 0.916 | 1.915 | 2.507 | 16 |
| | | 2 | 0.916 | 2.778 | 0.902 | 2.833 | 2.280 | 18 |
| | | 3 | 0.850 | 3.257 | 0.822 | 3.052 | 1.339 | 18 |
| | | 4 | 0.811 | 3.479 | 0.780 | 3.104 | 1.124 | 20 |
| | | 5 | 0.675 | 4.524 | 0.266 | 5.560 | 0.849 | 16 |

**Notes.**

LVs is the number of latent variables, and HLNs is the number of neurons in the hidden layer.

In summary, the wavelet coefficient modeling results of the mexh wavelet basis function were superior to those of the gaus2 wavelet basis function when using the same modeling method and the same decomposition scale. Moreover, the performance of the ELM model was better than the PLSR model when using the same decomposition scale of the same wavelet basis function. The inversion results of the feature wavelet coefficients extracted from the mexh wavelet transformation were better than those of the full-band wavelet coefficients. Therefore, the best modeling results were obtained when using wavelet coefficients obtained from CWT with the mexh wavelet basis function under a decomposition scale of 1 and feature wavelet coefficients extracted by CARS and then modeled by ELM (CWT(mexh_1)-CARS-ELM). The $R^2$, RMSE, and RPD of CWT(mexh_1)-CARS-ELM were 0.946, 1.408%, and 3.759, respectively. This result also demonstrated that the feature wavelet coefficients extracted by CARS helped improve modeling accuracy and reduce inversion errors.

**Table 3** Modeling results based on extracted feature band wavelet coefficients.

| Modeling method | Wavelet basis function | Scale | Training dataset | | Validation dataset | | | LVs/HLNs |
|---|---|---|---|---|---|---|---|---|
| | | | R² | RMSE(%) | R² | RMSE(%) | RPD | |
| PLSR | mexh | 1 | 0.946 | 1.929 | 0.866 | 2.443 | 2.209 | 3 |
| | | 2 | 0.898 | 2.652 | 0.823 | 1.976 | 1.999 | 3 |
| | | 3 | 0.879 | 2.905 | 0.836 | 2.986 | 1.736 | 4 |
| | | 4 | 0.855 | 3.168 | 0.809 | 2.838 | 2.035 | 4 |
| | | 5 | 0.785 | 3.853 | 0.719 | 3.584 | 1.684 | 4 |
| | gaus2 | 1 | 0.785 | 3.843 | 0.755 | 3.091 | 1.909 | 4 |
| | | 2 | 0.745 | 4.189 | 0.603 | 3.938 | 1.559 | 3 |
| | | 3 | 0.495 | 5.802 | 0.422 | 5.238 | 1.238 | 3 |
| | | 4 | 0.543 | 5.518 | 0.298 | 5.523 | 1.190 | 3 |
| | | 5 | 0.297 | 6.723 | 0.055 | 6.885 | 1.006 | 2 |
| ELM | mexh | 1 | 0.951 | 2.193 | 0.946 | 1.408 | 3.759 | 9 |
| | | 2 | 0.943 | 2.454 | 0.939 | 1.936 | 3.259 | 9 |
| | | 3 | 0.929 | 2.786 | 0.919 | 2.149 | 2.799 | 8 |
| | | 4 | 0.912 | 2.883 | 0.904 | 1.856 | 3.443 | 9 |
| | | 5 | 0.908 | 3.035 | 0.881 | 2.506 | 2.610 | 10 |
| | gaus2 | 1 | 0.932 | 2.672 | 0.927 | 1.825 | 3.413 | 8 |
| | | 2 | 0.912 | 2.940 | 0.906 | 1.717 | 3.537 | 9 |
| | | 3 | 0.846 | 3.553 | 0.836 | 2.281 | 2.606 | 9 |
| | | 4 | 0.837 | 3.634 | 0.811 | 3.269 | 1.591 | 9 |
| | | 5 | 0.562 | 5.317 | 0.110 | 6.372 | 0.762 | 10 |

# DISCUSSION

## Correlation of soil water content and wavelet coefficient

Wavelet transform is a method for spectral data processing with high resolution and adaptability that is commonly used in processing various types of noise signals (*Chen & Qian, 2011*). A previous study identified the wavelet basis function "Haar" as the best in estimating soil water content considering the structural trend and detailed features of multi-dimensional space of fusion data sources at different spatial scales (*Jiang et al., 2023*). Gaus4, which is similar to the absorption characteristics of the soil reflectance spectrum, has been shown to be an effective wavelet mother function when estimating the content of other components in soil based on hyperspectral data, such as zinc content in coal mine soil and organic matter in soil (*Yu et al., 2016a*; *Guo et al., 2022*). The present study used both the mexh and gaus2 wavelet basis functions, leading to wavelet coefficients that had a higher correlation with soil water content compared to those obtained from other wavelet basis functions (File S1). Mexh and gaus2 were also able to effectively suppress interference and highlight local information.

Different wavelet basis functions yield varying results and decomposition results also vary with different decomposition scales (*Zhou et al., 2020*). In this study, the correlation between the wavelet coefficients of mexh and gaus2 and the soil water content decreased

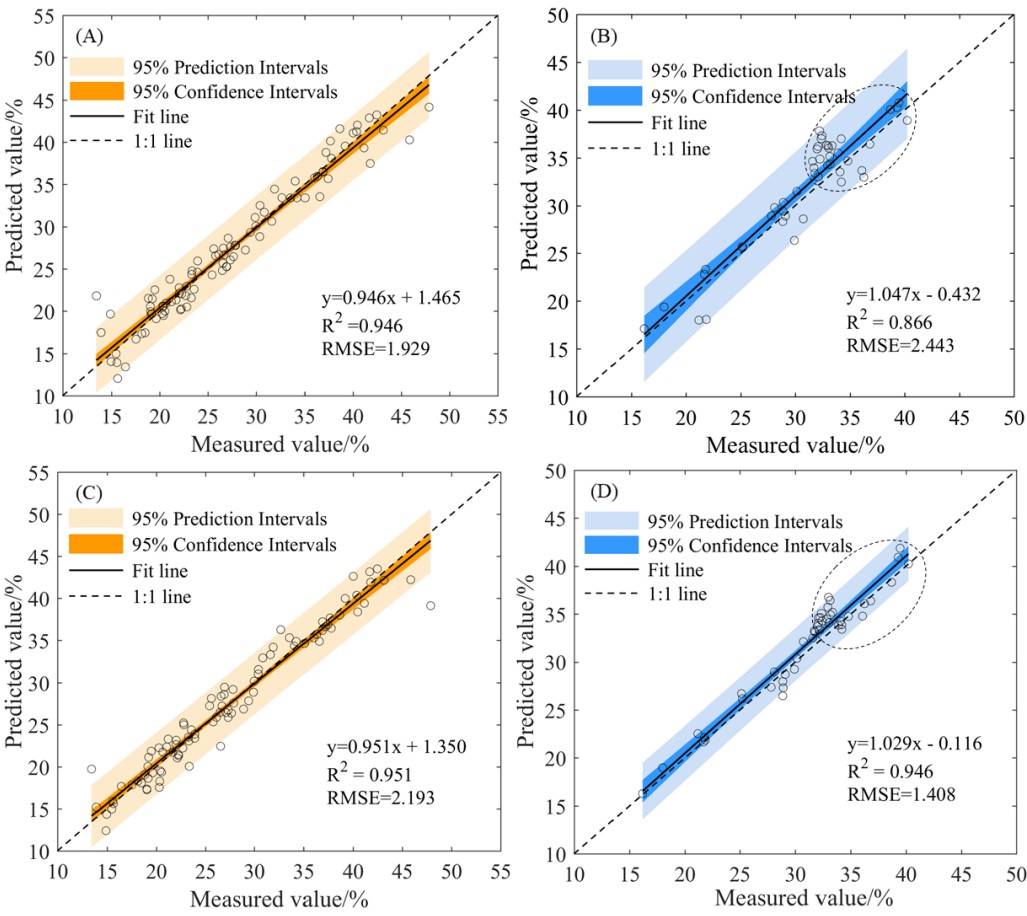

**Figure 7 The scatter plots of measured and predicted values by the PLSR and ELM models based on the feature wavelet coefficients.** (A) Training dataset (mexh-PLSR), (B) validation dataset (mexh-PLSR), (C) training dataset (mexh-ELM), and (D) validation dataset (mexh-ELM).

as the decomposition scale increased (Fig. 3). Over-decomposition occurred when the decomposition scale exceeded 8, while good correlation ($|r|>0.5$) was observed within decomposition scales of 1–5. Therefore, the decomposition scale 1–5 was selected, with the highly correlated bands mainly concentrated in the near-infrared region. For the remote sensing inversion of soil organic carbon, the original spectral reflectance and the first-order differential processed spectra processed by CWT within decomposition scales of 1–5 showed strong correlations with the target index, with a maximum correlation coefficient of 0.58 (*Jiang, Li & Yang, 2023*). In two previous inversion studies of soil moisture and soil organic carbon, the correlation gradually weakened as the wavelet decomposition scale increased, and the highly correlated bands were mainly concentrated in the near-infrared region (*Zhang et al., 2016*; *Gu et al., 2019*). Similar results were obtained in the present study. In another study, when CWT decomposed the hyperspectral data of winter wheat, the correlation between chlorophyll content and wavelet coefficients was first stronger and
then weakener within the decomposition scale of 1–9 (*Li et al., 2021a*). This may be due to variations caused by the different spectrums of research objects.

## Spectral sensitive bands of soil water content

CWT is able to effectively remove noise information and enhance local information when decomposing, but the amount of spectra is not reduced, and information might overlap. This study used the CARS method to extract sensitive wavelet coefficients. This method reduces redundant information and accurately and effectively determines the wavelength range of the feature wavelet coefficients (*Tang et al., 2021*; *Tang et al., 2023*). In this study, the feature wavelet coefficients decomposed by the mexh and gaus2 wavelet basis functions were mainly concentrated in the wavelength ranges of 931–967 nm and 1,015–1,184 nm, which belong to the near-infrared band. When hyperspectral remote sensing was used to extract soil moisture information, the wavelength bands extracted by linear forward stepwise regression were 450 nm, 574 nm, 986 nm, 1,400 nm, 1,672 nm, 1,998 nm, and 2,189 nm. These seven wavelengths were evenly distributed in the spectral domain and reflect the correlation between reflectivity and soil water content (*Liu et al., 2002a*). In a previous study measuring soil water content, when soil organic matter and iron content were low, there were two weak water absorption valleys near 1,200 nm and 1,770 nm; when the soil organic matter was low and the iron content was medium, there was a strong water absorption valley near 980 nm (*Stoner & Baumgardner, 1981*). A separate study that retrieved soil water content through remote sensing used the envelope elimination method to identify the sensitive bands for soil water content as 1,481 and 1,616 nm (*Bai et al., 2003*). Another study found that when the sensitive bands of the raw spectra of soil water content are mainly concentrated in the wavelength intervals of 1,400–1,600 nm and 1,900–2,300 nm, the sensitive bands of the first-order and second-order differential spectral indices by the one-dimensional, two-dimensional, and three-dimensional correlation matrix analyses are mainly concentrated in the intervals of 1,400, 1,900, and 2,100 nm (*Zhou, Zhou & Lao, 2021*). The results obtained in the present study are similar to those of previous studies. The characteristic wavelengths of soil information extracted by spectral remote sensing were all in the water absorption band region.

## The effectiveness of wavelet basis functions and establishment of models

Tables 2 and 3 show that the modeling results based on the characteristic wavelet coefficients of the mexh wavelet basis function were generally better than those of the gaus2. Because the mexh wavelet basis function is the negative second derivative of the Gaussian function, which has good localization in both the time and frequency domains, there is no scale function and no orthogonal. Therefore, compared to the gaus2 wavelet basis function, mexh has better detection and highlighting capabilities (*Deng et al., 2020*). The modeling results of the feature wavelet coefficients were generally better than the full-band wavelet coefficients because the process of variable extraction reduced the influence of low-correlation bands on the modeling results (*Xia et al., 2023*). In this study, the spectral data were decomposed and analyzed within decomposition scales of 1–10, but only the wavelet

coefficients of a decomposition scale of 1 were finally applied. Future studies should explore using multi-scale information.

When establishing the soil water content estimation model based on different datasets, the performance of the non-linear (ELM) method was superior to the performance of the linear (PLSR) model (Table 3)—a finding similar to previously reported results (*Guo et al., 2021*; *Wang et al., 2022*). This superiority was particularly evident in samples with soil water content higher than field capacity (Fig. 7), where the scatter of these points along the fitted line was more dispersed. Conversely, when the soil water content was lower than field capacity, the scatter plot was more concentrated. The non-linear method (ELM) has achieved satisfactory performance in estimating soil water content because of the non-linear relationship between soil water content and spectral reflectance (*Cai & Ding, 2018*; *More et al., 2022*). Because of the non-linear relationship between the spectral reflectance of soil with different levels of water content, prediction methods of soil water content above and below field capacity need to be further studied.

## Limitations and future research

Sandy clay loam was used as the research object in this study. Before the soil samples were prepared with different water content levels, the soil samples were ground and sieved. This pre-treatment made the particle size and organic matter of the soil samples similar and better captured the response of spectral reflectance to soil water. However, there are differences in soil structure and texture between the experimental soil used in this study and undisturbed soil in the field, so the hyperspectral retrieval model of soil water content identified in this study is not yet ready for practical application in field scenarios. Soil reflectance properties also vary greatly depending on the nature of the soil in different regions. For example, for soil water content of 10.13–34.83%, the spectral reflectance at a wavelength range of 350–649 nm was similar in both loess soil samples of northern Shaanxi and in the soil samples of the present study; however, for a soil water content of 4.01%-13.58%, the spectral reflectance at a wavelength range of 949–1,350 nm was higher in the loess soil samples of northern Shaanxi than in the soil samples of the present study (*Jiang et al., 2017*). While the soil water content of Heilongjiang black soil and Inner Mongolia Hetao irrigation soil were close to the soil water content of the present study, the spectral reflectance in the 400–1,400 nm band was higher in both Heilongjiang black soil and Inner Mongolia Hetao irrigation soil than the spectral reflectance of the soil samples in the present study (*Wang et al., 2023c*; *Wang et al., 2023b*). These examples demonstrate that the estimation of soil water content using hyperspectral data is influenced by soil type and components, resulting in significant variations in the relationship between soil spectral information and soil water content. Therefore, one specific soil estimation model cannot be directly applied to estimate soil water content in all scenarios.

The prediction model established in this study only considers the influence of soil water content on spectral reflectance. However, many factors in soil interfere with reflectance monitoring. There are also differences in the physicochemical and spectral properties of different types of soil. Combining feature extraction, vegetation index, and machine learning methods effectively improve target parameter prediction accuracy (*Tang et*

*al., 2021*). Salt also interferes with soil spectra, and filtering out the effects of salt in pretreatment using the external parameter orthotropic method can significantly improve model performance and achieve accurate soil water content estimation in salinized areas (*Peng et al., 2016*). Soil particle size and roughness also affect the stability and prediction accuracy of the model; classifying and predicting samples helps improve the prediction accuracy of target parameters (*Lu et al., 2018*). The influence of organic matter on soil spectra is closely related to changes in soil water content (*Shang et al., 2017*). The soil in this study were laboratory-prepared samples, which ignored the influencing factors in real environments; therefore, when applying the systematic method recommended in this paper to estimate soil water content in real scenarios, it is necessary to identify the soil type, soil surface roughness, and eliminate other factors that may interfere with soil water content estimation.

In summary, in order to better apply the method and results obtained in this study to real scenario, more complete soil information needs to be incorporated into the prediction model. Identifying the spectral characteristics of soil information, summarizing the law of spectral variation of soil in diverse environments, and expanding and perfecting the spectral database are the basis for improving the accuracy and applicability of soil information extraction models, as well as the prerequisites for precision agriculture research (*Yao et al., 2008*). The findings of this study will provide new ideas for estimating soil information by hyperspectral remote sensing.

## CONCLUSIONS

This study proposes a data processing method combining CWT with CARS to remove noise and extract variables. This study also explores the method for predicting soil water content based on hyperspectral data. The linear (PLSR) and non-linear (ELM) models were established based on the full-band and feature wavelet coefficients. The results demonstrate the following: (1) In the original spectra processed by CWT with mexh and gaus2 wavelet basis functions, the correlation between wavelet coefficients and soil water content decreased as the decomposition scale increased, and the strong correlation ($|r|>0.5$) appeared in the decomposition scale of 1–5. The mexh wavelet basis function maximized the correlation between soil water content and soil spectral information. (2) CARS reduced the number of variables involved in modeling; the number of feature wavelet coefficients ranged from three to 48, which reduced a lot of redundant information compared to the results when using full-band wavelet coefficients, and the corresponding bands of the extracted wavelet coefficients were mainly distributed in the near-infrared region. (3) The ELM model achieved a satisfactory accuracy based on the feature wavelet coefficients of the mexh wavelet basis function at a decomposition scale of 1, with an $R^2$ and RMSE of 0.951 and 2.193%, respectively, in the training dataset and an $R^2$, RMSE, and RPD of 0.946, 1.408%, and 3.759, respectively, in the validation dataset. Based on the results of this paper, the CWT(mexh_1)-CARS-ELM systematic modeling method is recommended for estimating the water content of sandy clay loam.

## ACKNOWLEDGEMENTS

The authors would like to thank Kailun Peng and Shuai Zhang of Kunming University of Science and Technology for their help with the experiment.

### Funding

This work was supported by the National Natural Science Foundation of China (52209056, 52009039), the Yunnan Fundamental Research Projects (202301AU070095, 202301AU070190), the Natural Science Research Foundation of Kunming University of Science and Technology, China (KKZ3202323004, KKZ3202323003), the Yunnan Province Undergraduate Innovation and Entrepreneurship Training Plan Program (S202210674096), the Yunnan Science and Technology Talent and Platform Program (202305AM070006), and the Scientific Research Fund Project of Yunnan Provincial Department of Education (2022J0061). The funders had no role in study design, data collection and analysis, decision to publish, or preparation of the manuscript.

### Grant Disclosures

The following grant information was disclosed by the authors:
National Natural Science Foundation of China: 52209056, 52009039.
Yunnan Fundamental Research Projects: 202301AU070095, 202301AU070190.
Natural Science Research Foundation of Kunming University of Science and Technology, China: KKZ3202323004, KKZ3202323003.
Yunnan Province Undergraduate Innovation and Entrepreneurship Training Plan Program: S202210674096.
Yunnan Science and Technology Talent and Platform Program: 202305AM070006.
Scientific Research Fund Project of Yunnan Provincial Department of Education: 2022J0061.

### Competing Interests

Lihua Luo is an employee of the Yunnan Institute of Water and Hydropower Engineering Investigation and Design, Co., LTD.

### Author Contributions

- Shaomin Chen conceived and designed the experiments, analyzed the data, prepared figures and/or tables, authored or reviewed drafts of the article, and approved the final draft.
- Jiachen Gao performed the experiments, analyzed the data, prepared figures and/or tables, authored or reviewed drafts of the article, and approved the final draft.
- Fangchuan Lou performed the experiments, prepared figures and/or tables, and approved the final draft.
- Yunfei Tuo conceived and designed the experiments, prepared figures and/or tables, and approved the final draft.

- Shuai Tan conceived and designed the experiments, analyzed the data, prepared figures and/or tables, and approved the final draft.
- Yuyang Shan conceived and designed the experiments, prepared figures and/or tables, and approved the final draft.
- Lihua Luo analyzed the data, authored or reviewed drafts of the article, and approved the final draft.
- Zhilin Xu performed the experiments, authored or reviewed drafts of the article, and approved the final draft.
- Zhengfu Zhang performed the experiments, authored or reviewed drafts of the article, and approved the final draft.
- Xiangyu Huang performed the experiments, authored or reviewed drafts of the article, and approved the final draft.

## Data Availability

The raw measurements are available in the Supplementary File.

## Supplemental Information

Supplemental information for this article can be found online at http://dx.doi.org/10.7717/peerj.17954#supplemental-information.

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
