# Peer review of "Rapid estimation of soil water content based on hyperspectral reflectance combined with continuous wavelet transform, feature extraction, and extreme learning machine"

_PeerJ, doi:10.7717/peerj.17954_

## Round 0.1 · original submission · Major Revisions

The manuscript addresses a very interesting subject. However, it needs substantial improvement before being published in PeerJ.

Please consider the very helpful comments by the reviewers.

In addition to the reviewers' comments, I am concerned about some methodological issues. Apparently, a single soil has been used, which has been brought to different moisture levels. No information is provided about this soil, particularly its texture and organic matter content. Furthermore, the soil is ground and sieved before use, which leads to conditions completely different from field conditions. Do you think that the results obtained in these conditions can be extrapolated to different soils in field conditions? It is necessary to justify it or indicate that the results are valid only for soils of a certain texture and a certain organic matter content.

The manuscript would benefit from revision by a fluent English speaker.

**Language Note:** The Academic Editor has identified that the English language must be improved. PeerJ can provide language editing services - please contact us at [email protected] for pricing (be sure to provide your manuscript number and title). Alternatively, you should make your own arrangements to improve the language quality and provide details in your response letter. – PeerJ Staff

Reviewer 1 ·

Basic reporting

1.The article provides a detailed description of the current research status and progress in using hyperspectral technology for soil moisture monitoring, and elaborates on the limitations of monitoring accuracy due to spectral data saturation. However, other factors that affect soil moisture monitoring accuracy are not mentioned in the entire article. It is recommended to increase the discussion of this issue.

Experimental design

2. The author provides a detailed description of the principle and calculation process of the Competitive Adaptive Re Weighted Sampling (CARS) method used in the article, but the advantages of this method in extracting wavelet coefficients and its applicability to water content monitoring are not clear enough.
3.The article provides a detailed description of the collection method and container specifications for soil samples, but lacks information on the characteristics of the experimental environment, such as temperature, humidity, soil irrigation, and other related data, which may also have an impact on the experimental results.

Validity of the findings

No comment.

Additional comments

This article proposes a quantitative estimation method for soil moisture content in hyperspectral remote sensing based on a combination of continuous wavelet transform (CWT) and competitive adaptive reweighted sampling (CARS). Soil moisture inversion models based on full band wavelet coefficients and feature band wavelet coefficients are established, and the accuracy of the models is evaluated to explore suitable wavelet coefficients and modeling methods, This provides technical support for establishing accurate and stable soil moisture estimation models. However, there are still the following issues that need improvement in terms of details:
1.The article provides a detailed description of the current research status and progress in using hyperspectral technology for soil moisture monitoring, and elaborates on the limitations of monitoring accuracy due to spectral data saturation. However, other factors that affect soil moisture monitoring accuracy are not mentioned in the entire article. It is recommended to increase the discussion of this issue.
2. The author provides a detailed description of the principle and calculation process of the Competitive Adaptive Re Weighted Sampling (CARS) method used in the article, but the advantages of this method in extracting wavelet coefficients and its applicability to water content monitoring are not clear enough.
3.The article provides a detailed description of the collection method and container specifications for soil samples, but lacks information on the characteristics of the experimental environment, such as temperature, humidity, soil irrigation, and other related data, which may also have an impact on the experimental results.
4. In the results and analysis, the model based on wavelet coefficients and soil moisture content selected the optimal wavelet coefficients under the same modeling method or decomposition level. However, the description of the feature data used in all models is not clear enough. It is necessary to describe in more detail which features are used to train the model and which data are used for validation.
The article should be accepted if the authors make some revisions, minor enough that I would NOT necessarily need to re-review it.

Reviewer 2 ·

Basic reporting

In this study, a soil water content estimation model was constructed by means of the spectral features of the continuous wavelet transform, and it well describes the superior performance of the method in estimating soil water content with satisfactory estimation accuracy.
1. The title highlights too much of the methodology used in the study, and all these words are also found in the keywords; it is recommended that the authors revise the title.
2. In this manuscript, the image caption descriptions are too simple; the image captions should stand alone, i.e., be sufficiently descriptive without having to refer to the body text; please revise further.

Experimental design

Line 315-316, What are the criteria for classifying the five gradients of soil moisture content?

Validity of the findings

1. This study chose 1–5 wavelet scales for soil water content estimation; why not choose the optimal wavelet features from these 5 scales for modeling? And I think there is strong collinearity between many wavelet features in hyperspectral information and whether the wavelet features screened using competitive adaptive screening at different scales eliminate the collinearity.
2. The discussion lacks a presentation of the study's shortcomings, and I suggest that the authors discuss this further.

Reviewer 3 ·

Basic reporting

I would like to thank the editor and authors for the opportunity to review this interesting study. In this paper, the author prepared a large range of soil water content samples, and obtained the hyperspectral reflectance. The most important is to process the spectral information by using wavelet transform, and extract the relevant variables by the CARS method. Finally, the linear and nonlinear prediction model is established, and the final recommended model has a satisfactory accuracy. Overall, this manuscript is a valuable and interesting piece of research, and I think this work is suitable for publication. The paper is well-written, well-organized, and substantial in its analysis of the topic.

Experimental design

No comment.

Validity of the findings

No comment.

Additional comments

However, there is still room for improvement. At the same time, I have a question. They should clarify to reduce doubts and make the article more reliable. The specific comments are attached:

Q1 Published papers on the prediction of soil water status seem to prefer to focus on the condition of low water content, which helps guide agricultural irrigation and ecological restoration monitoring. The authors of this paper predict the moisture content of soil samples that are higher than the field water capacity, and how this is considered.

Q2 Line 142: Why was 1.7cm selected as the limit thickness of the sample?

Q3 Lines: 226-227: Why choose these two models for establishing the inversion model? Is there a reason?

Q4 Lines: 268-269, what is the meaning of “L” and “M” in Equations (11) and (12), the relevant note is not seen in the context, please ask the author to explain.

Q5 Figure 5 (b) shows RMSEV as the ordinate, which seems to be RMSECV. Please check with the author.

Q6 What is “RMSEV” in Figure 5 (b)? Which seems to be “RMSECV”. What is the unit? Please add in the Figure.

Q7 In Figure 7, what is the purpose of the 1:1 line? The fitted lines in Figure 7 (a) and (b) overlap with 1:1 lines. Do they need to be extended appropriately?

Reviewer 4 ·

Basic reporting

See annex for details of comments

Experimental design

See annex for details of comments

Validity of the findings

See annex for details of comments

Additional comments

See annex for details of comments

Annotated reviews are not available for download in order to protect the identity of reviewers who chose to remain anonymous.

Reviewer 5 ·

Basic reporting

Please see the comments below together.

Experimental design

Please see the comments below together.

Validity of the findings

Please see the comments below together.

Additional comments

PeerJ Review Report: Rapid estimation of soil water content based on hyperspectral reflectance combined with continuous wavelet transform, competitive adaptive reweighted sampling and extreme learning machine
The manuscript presents a method for estimating soil water content using hyperspectral remote sensing, leveraging continuous wavelet transform (CWT), competitive adaptive reweighted sampling (CARS), and extreme learning machine (ELM). Although the study employs innovative techniques and offers a detailed methodology, there are several areas where significant improvements are required to meet the standards of PeerJ:
Originality and Contribution: The manuscript needs to more clearly articulate its novelty and contribution to the field. Comparative analysis with existing methods and a more thorough discussion of the methodological advancements are required.
Methodological Rigor and Validation: The validation of the proposed methods against ground-truth measurements or an established benchmark is essential but seems to be either insufficiently detailed or lacking robustness. Enhancing the validation section with more comparative data and statistical analysis would strengthen the manuscript.
Data Availability and Reproducibility: Ensuring that data and code used in the study are available for review and replication is crucial. The manuscript should explicitly address data availability and provide access to scripts or software used in the analysis.
Clarity and Organization: The manuscript would benefit from restructuring to improve flow and readability. Certain sections appear overly technical without sufficient explanation for a broader audience. Simplifying complex discussions and adding summaries or conclusions to dense sections could aid reader comprehension.
Literature Review: Expanding the literature review to more comprehensively cover related work would contextualize the study's contribution. Highlighting gaps in current research that the manuscript addresses would also clarify its significance.
Environmental and Practical Implications: The discussion could be expanded to include more on the practical applications of the research findings, particularly how they can be used in soil management practices and the broader environmental implications.
Figures and Tables: Ensure that all figures and tables are clearly labeled, high quality, and directly referenced within the text. Any supplementary material should be easily accessible and relevant.
Technical Details and Assumptions: Some sections are heavy with technical jargon and assumptions that may not be clear to all readers. Clarifying these aspects and explaining the rationale behind certain methodological choices would be beneficial.
Given these points, the manuscript may not currently align with PeerJ's focus on high-impact, interdisciplinary contributions to science. Major revisions in the areas highlighted above could improve the manuscript significantly, making it more suitable for publication in a journal that closely matches its scope and contribution to the field.

---

## Round 0.2 · Major Revisions

Although two reviewers recommend accepting the revised manuscript, reviewer 5 is not satisfied with the revision and again requests major revisions. Please follow his recommendations.

As for my previous suggestions, I am not satisfied with the revision either. The authors have added some information on soil properties. The influence of soil properties is referred to in lines 515-526, in response to reviewer 1. However, I miss a justification on:

- Whether a method derived from a single soil (silt loam) can be extrapolated to a variety of soils.
- Whether a method derived from results obtained in the laboratory from a ground and sieved soil (destroying soil structure and modifying soil texture) can be extrapolated to real soils in field conditions.
Which information can be obtained from your study to be applied in a real scenario? This must be clarified in the objectives and justification as well as in the conclusions.

Additional comments:
- Lines 303-310 are not results but Methods.
- What does “the same as below” in some Table titles mean?
Moreover, I insist that the English language must be revised by a fluent English speaker.

Therefore, I recommend again revisions.

Reviewer 3 ·

Basic reporting

The author has completed all revisions as required and meets the requirements of the journal. It is recommended to accept.

Experimental design

No comment.

Validity of the findings

No comment.

Additional comments

No comment.

Reviewer 4 ·

Basic reporting

I think the author has carefully revised the manuscript and it is now ready for acceptance.

Experimental design

I think the author has carefully revised the manuscript and it is now ready for acceptance.

Validity of the findings

I think the author has carefully revised the manuscript and it is now ready for acceptance.

Reviewer 5 ·

Basic reporting

I think authors have revised the manuscript well, but many comments on discussion and quality improvement remained unaddressed.

Experimental design

Clarity still needs improvement.

Validity of the findings

Need to address the comments well.

Additional comments

While authors have revised the manuscript in quite detail, but majority of them are in editorial. There are some concerns about the science and in-depth analysis of the data, which must be improved.

---

## Round 0.3 · Minor Revisions

The concerns of reviewer 5 have been addressed, so that they recommend accepting the manuscript.

The authors have more or less responded to my comments. However, I still have some concerns:

Line 515: You identify the soil as “red soil”. This is not informative for the common reader. Please classify the soil studied according to the WRB or the Soil Taxonomy. In addition, specify the geographical origin of the soil. As follows: “The soil used in this study was a … from …”

Lines 518-519: “there are differences in soil structure and composition between the experimental soil used in this study and undisturbed soil in the field”. Differences are expected in structure and texture, not so much in composition.

Line 545: “the soil type (texture)”. The texture alone does not define the soil type. The soil type is defined by a number of diagnostic soil properties and horizons.

Lines 569-570: “red soil”. Please see my comments to line 515.

My concern that you number as 3 has not been satisfactorily addressed. In the objectives and justification, it must be clarified that you used a soil that has been modified by grinding and sieving, and you must specify to what extent you expect that the results obtained for that soil can be extrapolated to a soil in a real scenario. The same for the conclusions, where you only refer to "red soil", which is not informative at all.

These are the reasons why I consider that the manuscript still needs minor revisions.

Reviewer 5 ·

Basic reporting

I think the authors have revised the manuscript carefully and I addressed my comments.

Experimental design

No Comment

Validity of the findings

No Comment

Additional comments

I think the authors have revised the manuscript carefully and I addressed my comments.

---

## Round 0.4 · accepted · Accept

The authors have made a thorough revision of the manuscript since its first version, following the recommendations of the reviewers and my own. Therefore, I consider that the manuscript in its current form is suitable for publication in

PeerJ.